# Development and Evaluation of an FDM Printed Nasal Device for CPZ Solid Nanoparticles

**DOI:** 10.3390/molecules28114406

**Published:** 2023-05-29

**Authors:** Thinh To Quoc, Krisztina Bíró, Ágota Pető, Dóra Kósa, Dávid Sinka, István Lekli, Attila Kiss-Szikszai, István Budai, Mónika Béres, Miklós Vecsernyés, Pálma Fehér, Ildikó Bácskay, Zoltán Ujhelyi

**Affiliations:** 1Department of Pharmaceutical Technology, Faculty of Pharmacy, University of Debrecen, Nagyerdei körút 98, 4032 Debrecen, Hungary; toquocthinh_93@yahoo.com.au (T.T.Q.); peto.agota@pharm.unideb.hu (Á.P.); kosa.dora@pharm.unideb.hu (D.K.); sinka.david@pharm.unideb.hu (D.S.); vecsernyes.miklos@pharm.unideb.hu (M.V.); feher.palma@pharm.unideb.hu (P.F.); bacskay.ildiko@pharm.unideb.hu (I.B.); 2Doctoral School of Pharmaceutical Sciences, University of Debrecen, Nagyerdei körút 98, 4032 Debrecen, Hungary; biro.krisztina@med.unideb.hu; 3Institute of Healthcare Industry, University of Debrecen, Nagyerdei körút 98, 4032 Debrecen, Hungary; 4Hospital Pharmacy at the University of Debrecen, University of Debrecen, Nagyerdei körút 98, 4032 Debrecen, Hungary; 5Department of Pharmacology, Faculty of Pharmacy, University of Debrecen, Nagyerdei körút 98, 4032 Debrecen, Hungary; lekli.istvan@pharm.unideb.hu; 6Department of Organic Chemistry, University of Debrecen, Egyetem tér 1, 4010 Debrecen, Hungary; kiss.attila@science.unideb.hu; 7Department of Engineering Management and Enterprise, Institute of Industrial Process Management, Faculty of Engineering, University of Debrecen, Ótemető utca 2, 4028 Debrecen, Hungary; budai.istvan@eng.unideb.hu; 8Department of Medical Imaging, Department of Radiology and Imaging Science, Faculty of General Medicine, University of Debrecen, Nagyerdei körút 98, 4032 Debrecen, Hungary; beres.monika@med.unideb.hu

**Keywords:** chlorpromazine, nanostructures, penetration enhancers, spray drying, cytotoxicity investigation, MTT test, TEER, RPMI cells

## Abstract

Nasal drug delivery has been a focus of scientific interest for decades. A number of drug delivery systems and devices are available and have been highly successful in providing better and more comfortable therapy. The benefits of nasal drug delivery are not in question. The nasal surface provides an excellent context for the targeted delivery of active substances. In addition to the large nasal surface area and intensive absorption, the active substances delivered through the nose overcome the blood–brain barrier and can be delivered directly to the central nervous system. Formulations for nasal administration are typically solutions or liquid dispersed systems such as emulsions or suspensions. Formulation techniques for nanostructures have recently undergone intensive development. Solid-phase heterogeneous dispersed systems represent a new direction in pharmaceutical formulations. The wide range of possible examples and the variety of excipients allow for the delivery of a wide range of active ingredients. The aim of our experimental work was to develop a solid drug delivery system that possesses all of the above-mentioned advantageous properties. In developing solid nanosystems, we not only exploited the advantages of size but also the adhesive and penetration-enhancing properties of excipients. During formulation, several amphiphilic compounds with adhesion properties and penetration enhancing effects were incorporated. We used chlorpromazine (CPZ), which is mainly used in the treatment of psychotic disorders such as schizophrenia and bipolar disorder. Chlorpromazine has been previously investigated by our team in other projects. With the availability of previous methods, the analytical characterization of the drug was carried out effectively. Due to the frequent and severe side effects of the drug, the need for therapeutic dose reduction is indisputable. In this series of experiments, we succeeded in constructing drug delivery systems. Finely divided Na nanoparticles were formed using a Büchi B90 nanospray dryer. An important step in the development of the drug carrier was the selection of suitable inert carrier compounds. Particle size determination and particle size distribution analysis were performed to characterize the prepared nanostructures. As safety is the most important aspect of any drug formulation, all components and systems were tested with different biocompatibility assays. The tests performed demonstrated the safe applicability of our systems. The bioavailability of chlorpromazine was studied as a function of the ratio of the active ingredient administered nasally and intravenously. As described above, most nasal formulations are liquids, but our system is solid, so there is currently no tool available to accurately target this system. As a supplement of the project, a nasal dosing device was developed, corresponding to the anatomical structure; a prototype of the device was made using 3D FDM technology. Our results lay the foundation for the design and industrial scaling of a new approach to the design and production of a high-bioavailability nasal medicinal product.

## 1. Introduction

Nasal drug administration has been a focus of scientific interest for decades, since the beneficial properties of the nasal mucosa were discovered. It not only provides a large absorption surface for drug absorption, with a rapid effect achieved, but an alternative route is made possible by bypassing the blood–brain barrier to the central nervous system also [1]. Moreover, the first-pass metabolism of pharmacons in the liver can be reduced [2]. Studies have demonstrated that many types of active substance can be introduced nasally into the systemic circulation, even large molecules such as peptides and proteins, especially in the presence of permeability-enhancing excipients. The number and the importance of nasally administered drugs with systemic effects are increasing [3]. The application of these formulations seems most beneficial in case of diseases that require immediate and rapid treatment. Preparations intended for nasal use are typically solutions or liquid-dispersed systems, such as emulsions or suspensions [4]. The preparation of these traditional dosage forms is relatively simple and the stability of these systems is excellent; however, patients report complaints during their use [5]. Specifically, patients report the liquid dripping into the pharynx and the resulting unpleasant taste. From a pharmaceutical point of view, the loss of active ingredients due to their short duration constitutes a problem. These properties can be improved by using suitable auxiliary materials with appropriate adhesive properties [6]. The objective of this research was to formulate and investigate novel solid lipid nanoparticles with enhanced API penetration. In addition to the API formulation, a novel nasal dispenser was designed, manufactured, and tested to complete the investigation. Chlorpromazine (CPZ), an antagonist of D2 dopamine receptors and similar receptors, such as D3 and D5, is primarily used to treat psychotic disorders including schizophrenia and bipolar disorder [7]. In this study, chlorpromazine was selected as the active pharmaceutical component due to its well-characterized analytics and the need for decreased therapeutic doses due to frequent and serious side effects. Common side effects include movement problems, sleepiness, dry mouth, low blood pressure upon standing, and increased weight [8]. Serious side effects may include the potentially permanent movement disorder tardive dyskinesia, neuroleptic malignant syndrome, severe lowering of the seizure threshold, and low white blood cell levels. In older people with psychosis as a result of dementia, it may increase the risk of death. It is unclear whether it is safe for use in pregnancy [9]. As modern therapeutic systems, nanoparticles have brought about a completely new era in the field of drug delivery; they deliver the active pharmaceutical compound to the tissues and cells in an unchanged and concentrated form, minimizing the systemic effects and toxicity to a large extent [10]. This possibility has opened up new horizons in the diagnosis and treatment of human diseases. The areas in which nanoparticles are used are highly diverse, as they provide a great deal of help in the organization of therapies, so they most likely provide or can provide a solution for the treatment and diagnosis of the most serious diseases (i.e., tumors) that occur today [11]. Biodegradable nanoparticles can be produced from polylactic acid (PLA), polylactic acid-glycolic acid (PLGA), and polymethyl methacrylate (PMMA) [12]. The polymer–drug conjugation makes the targeted therapy more effective, especially if penetration-enhancing amphiphilic compounds are also used as excipients. Surface-active agents are widely used in the development of new pharmaceutical dosage forms to improve the bioavailability of drugs that have low solubility in water [13]. They can influence drug permeability by modifying barriers, and by micellar solubilization, membrane fluidization, ion-pair formation, and the inhibition of efflux transporters such as P-glycoprotein. On the other hand, surfactants can cause local irritation, membrane damage, and cell death; therefore, during formulation processes, in vitro tests are required for the investigation of cytotoxicity and irritative effects [14]. According to these observations, a combination of penetration enhancers and polymers might be a promising tool to improve the utilization of active ingredients [15]. The formation of self-assembling emulsion systems was achieved by the titrimetric dilution of the appropriate combination of surfactants. Since safety is the most important aspect of all pharmaceutical formulations, all components and systems were tested with biocompatibility tests. Biocompatibility assays of the selected compounds and performed systems were performed on the RPMI 2650 immortalized nasal epithelial cell line by an MTT viability assay [16]. We found proof of biocompatibility as an essential milestone for further development. In pharmaceutical technological development, the production of nano-sized carrier systems is currently the most researched area [17]. In our experiments, the selection of the applied formulation technique was based on reliability. Several well-defined methods are available for the production of solid nanoformulations. The nanostructure can be created using simple methods such as spray-drying technology. This process was an intensive area of pharmaceutical technological development in the near past, ensuring the high efficiency and reliability of these methods [18]. Solid-phase nanoparticles were formed using Büchi Nano Spray Dryer B-90 HP apparatus (Büchi Labortechnik AG; Flawil, Switzerland). B-90 HP is well-defined equipment for nanoparticle formulation at the laboratory scale. Nanostructures can be produced with a Nano Spray Dryer device in four general steps: sample preparation, the atomization and drying of the droplets within the drying chamber, the capturing of the produced particles by the collecting electrode, and the recovery of the powder [19]. The physical properties of the formed systems were examined using two methods. The particle size distribution of the formed nanoparticles was investigated in the dispersion state using a Malvern Nano Zetasizer ZSP (Malvern Panalytical; Malvern, UK) [20]. Scanning electron microscope (SEM) images were taken to examine the morphology of the formed nanoparticles. The SEM investigations confirmed the appropriate morphology of the particles [21]. Size distribution measurements confirmed that the size of the produced particles is between 30 and 300 nm for the different compositions. The administration of finely divided nano-powders requires the experimental development of a special dosing device. Nasal drug delivery devices were manufactured using FDM 3D printing technology based on a 3D computer design of the anatomical features. The produced devices were refined based on the generous feedback from our volunteers. Our results might provide useful data for further development strategies of nasal drug delivery systems.

## 2. Results

### 2.1. Formulation of Self-Assembling Emulsion Systems

The formulation of the nanostructures started with the development of self-organizing heterogeneous dispersed systems from CPZ and the selected excipients. The aim of constructing the pseudoternary phase diagram was to determine the occurrence range of self-organizing combinations. The pseudoternary phase diagram of combinations is presented in Figure 1. Nanoemulsions formed with various proportions of components, mainly during the titration. Five different compositions were selected, according to the phase diagram. These compositions are able to form self-organizing systems. Our systems were stable and clear after four weeks of standing. The compositions are shown in Table 1.

### 2.2. Setting the Formulation Parameters

The spray-drying technique is frequently used for solid nanoparticle formulations; in our experiments, many manufacturing parameters were found to be crucial to ensuring the required product properties. In our first attempts to prepare spray-dried nanoparticles, a significant loss of CPZ was measured. Results are presented in Figure 2. In these formulations, the following parameters were set up. The temperature of the inlet gas was 100 °C, which decreased to 33 °C when it left the formulation chamber. During the formulation, the flow rate was 0.11 m^3^/h and the chamber pressure was 31 mbar. In addition to the complete exclusion of light, the inlet temperature was decreased to 60 °C while the flow rate was increased to 0.16 m^3^/h and the chamber pressure was 50 mbar, but the output temperature was still measured as 30 °C. The feed pump rate (90% of full operation speed), nebulizer rate (80% of full operation), and nebulizer voltage (122 kHz) were not altered.

### 2.3. Evaluation of the Size and Size Distribution of Dispersed Droplets

The size distribution measurements confirmed that the compositions constituted heterogeneous dispersed systems in the 10–900 nm ratio. The evaluated size and distribution were as follows: Composition 1; d = 25–90 ± 5 nm, Composition 2; d = 150–950 ± 5 nm, Composition 3; d = 95–250 ± 5 nm, Composition 4; d = 25–110 ± 5 nm, Composition 5; d = 10–40 ± 5 nm. Results demonstrated in Figure 3.

### 2.4. Evaluation of the Size and Morphology of Solid Nanostructures

Scanning electron microscope examinations confirmed the designed regular spherical shape. The morphology of the particles had been demonstrated in Figure 4. The size distribution measurements confirmed that the compositions formed heterogeneous dispersed systems in the 10–900 nm ratio.

### 2.5. Investigation of the Distribution and Aggregation Properties of Solid Nanoparticles in Primary Containers

During the investigation, we examined the degree of aggregation that can achieved by placing the nanosystems in primary packaging. Our observation is that higher HLB numbers of the applied surfactants can decrease the aggregation rate. Observations are demonstrated in Figure 5.

### 2.6. Investigation of the Cytotoxic Effect of Components and Their Blends on RPMI 2650 Nasal Epithelial Cells Using an MTT Viability Assay

The evaluated viability results demonstrated that the applied excipients and their blends did not significantly decrease the cell viability. A minor cytotoxic effect was detected at higher concentrations than those used in these applications. None of the determined values were close to the IC_50_ values. Morphology of RPMI 2650 was demonstrated in Figure 6, cell viability measurements had been summarized in Figure 7 and Figure 8.

### 2.7. Dissolution of CPZ from the Formulated Nanocarriers

The dissolution measurements demonstrated the rapid release of the incorporated API. After 15 s, more than 60% of the API from each composition was detected. Evaluation of dissoluted amount of CPZ was demonstrated in Figure 9. The total amount of CPZ was detected after 45 s, and the dissolution ratio did not change after 2 min.

### 2.8. Development and Pilot Production of a Medical Device for Nasal Delivery Systems

After testing the prototype on 50 human volunteers of random ages and sexes, the general shape EGG of the NF was well received. The NF fits well into the nostril, meaning that the design is effective. In the event of sneezing, the NF does not fly off. Female volunteers especially preferred using the BEL model due to its aesthetic modification capabilities, e.g., raising the nose line, even when this created drawbacks in terms of convenience and efficiency. The BEL model has a specific orientation fit, unlike the completely symmetrical EGG design; it also has a smaller air exchange surface when compared to an EGG prototype of equal size. The 3D-printed prototypes made from TPU did not cause allergic reactions in the volunteers unless the crescent blades’ glue was not cured sufficiently. There was one case of layer separation at the contact point of the bridge and barrel due to excess force being exerted and prolonged usage; the NIF broke off, but no harm was caused to the tester. Full working prototypes were given out to eight volunteers of random ages and sexes with a known history of dust allergies; five volunteers reported their symptoms, and three did not give feedback. Four out of five volunteers reported favorable results when using the NIF with a G3 filter; allergic symptoms decreased noticeably. In conclusion, the 3D-printed prototypes were highly useful during the pilot phase when determining the optimized shape and as a proof of concept. The NIF prototype was prepared in 1 h and ready to use after 24 h of curing; 1 person can craft 5 pieces per day, on average, using the 3D FDM technique.

## 3. Discussion

Medical products for nasal administration are highly significant nowadays [22]. Special attention is paid to nanostructures as possible drug carriers. They possess many beneficial properties, but their application requires attention. The techniques for formulating nanosturctures have undergone intensive development in recent years [23]. Solid-phase heterogeneous dispersed systems represent a new direction in pharmaceutical technology formulations. Numerous implementation options and the diversity of auxiliary materials enable the delivery of a wide variety of active ingredients. The aim of the present research was to form solid nano drug carrier systems for nasal application. During the excipient selection, amphiphilic compounds with precellular transport effects were shown to be preferable [24]. In our study, we demonstrated that self-organizing systems can be created from the selected amphiphilic auxiliary materials, which are suitable for nano spray-drying technology. As result of titrations, five different combinations of self-organizing heterogenous dispersed systems were produced. However, the spray-drying technique is a method frequently used for solid nanoparticle formulations. In our experiments, many manufacturing parameters were found to be crucial to ensuring that the product had the required properties. Formulation speeds and the temperature adjustment of the inlet and output gas flow and, as well as the complete exclusion of light, were found to be essential. The proper adjustment of these parameters resulted in dramatic improvements in API stability. Since biocompatibility means that safety is an essential aspect of pharmaceutical development, in vitro cell viability tests were performed on RPMI 2650 cells. The RPMI 2650 cell culture is a reliable in vitro model for intra-nasal administration [25]. The biocompatibility assays we performed demonstrated that the applied excipients are non-toxic and can also be used safely in combinations. During the physical tests of the systems, we confirmed that they were in the desired size range and that their morphology met the required standards. The stability test showed that the systems are stable, so they can be used as nasally administered API delivery systems. The dissolution test confirmed that the rapid release of CPZ was ensured and that the API remained stable in the formulation. The possibility of the excessive aggregation of the manufactured CPZ nanoparticles was excluded by microCT analysis [26]. The appropriate administration of the systems was ensured with the specially developed nasal dosing device, created using 3D technology. The device was proven to be suitable for delivering the systems to the target. Our results can serve as a useful basis for producing subsequent nasal or oral nanosystems, developing test methods, and designing biocompatibility tests.

## 4. Materials and Methods

### 4.1. Materials

Chlorpromazine, Kolliphor EL, Miglyol 840, dimethyl sulfoxide, glycerol, propylene glycol, polyvinyl alcohol, and bovine serum albumin were purchased from Sigma-Aldrich (St. Louis, MI, USA). Lauroglycol FCC, Transcutol HP, Labrafil 1944, and Labrasol were gifted by Gattefossé (Lyon, France). The human nasal epithelial cell line (RPMI 2650) was from the American Type Culture Collection (ATCC, Manassas, Virginia, USA). The MTT reagent 3-(4,5-Dimethylthiazol-2-yl)-2,5-diphenyltetrazolium bromide, and the buffer solutions, such as Hank’s Balanced Salt Solution (HBSS) and phosphate-buffered saline (PBS), were purchased from Sigma-Aldrich (St. Louis, MI, USA). The RPMI cell culture maintenance medium solution, TrypLE™ Express Enzyme (no phenol red), was ordered from Thermo Fisher Scientific (Waltham, MA, USA). Ninety-six-well cell plates and culturing flasks were obtained from VWR International (Debrecen, Hungary).

### 4.2. Formulation of SNEDD Systems

The formulation of self-assembling emulsion systems was performed by the titrimetric dilution of the appropriate combination of surfactants (Figure 1) [27]. In each composition, 10 mg CPZ was incorporated. The compositions of rgw excipients are listed in Table 1. To evaluate any signs of phase separation, the mixtures were observed for 24 h.

### 4.3. Formulation of Solid Nano Carriers

Solid-phase nanoparticles were formed using Büchi Nano Spray Dryer B-90 HP apparatus [28]. SNEDDS were added to 1% PVA solution and it was pumped into the equipment with a 90% pump rate. The drying gas was heated, and the inlet temperature and outlet temperature were 100 °C and 33 °C, respectively. The inlet temperature was decreased to 60 °C in the second manufacturing process and the flow rate was altered from 0.11 m^3^/h to 0.16 m^3^/h. The product was collected using a special rubber spatula. The actuator was driven at 122 kHz. The physical properties of the formed systems were examined using two methods. The particle size distribution of the formed nanoparticles was investigated in the dispersion state using a Malvern ZSP Nano Zetasizer (Malvern Panalytical; Malvern, UK).

### 4.4. Scanning Electron Microscopy

Electron microscopic analysis of the size, shape, and surface area of the formulated nanoparticles was performed using a Hitachi Tabletop microscope (TM3030 Plus) (Hitachi High-Technologies Corporation, Tokyo, Japan) [29]. The samples were attached to a plate covered with double-sided adhesive tape. An accelerating voltage of 5 kV was used during micrography.

### 4.5. Determination of Droplet Size

To determine the particle size of the formulated nanoparticles, a dynamic light-scattering device (Malvern ZSP Nano Zetasizer, Malvern Panalytical; Malvern, UK) was used [30]. Half a gram of the samples was dissolved in 100 mL distilled water and was exposed to a monochrome light wave. When this light meets a solution containing macromolecules, the light is scattered in all directions.

### 4.6. Fused Deposition Modeling

FDM was performed using CRAFTBOT 3 (Craftbot Ltd., Budapest, Hungary) equipment [31]. Direct drive independent extruders (IDEX) were used. The printing size was 374 × 250 × 250 mm, the layer resolution was adjusted to 50–300 microns, the positioning precision (XY) was 4 microns, Z: 2 microns, the filament diameter was 1.75 mm, the selected nozzle had a 0.25–0.8 mm diameter, and the nozzle temperature was measured as 180–300 °C. The bed temperature was set to 50–110 °C, and the print speed was 50–200 mm/s. For the USB host and Wi-Fi IF, Win7/8/10, OSX, and Linux, see the Supplementary Materials: PLA, ABS, PVA, PET, HIPS, NYLON, etc. MDFlex Copper3D (Santiago, Chile) antibacterial filaments were used. MDFlex is an innovative nanocomposite developed with a high-quality TPU98A and a patented nano-copper additive; it has been scientifically validated and is highly effective. To create the FDM-ready item design, Craftbot 3 Slicer Craftware Pro Premium 3.1. Software was used. The slicer settings were as follows: nozzle size, 0.4 mm; resolution extrusion width, 0.400 mm; layer height, 0.100 mm; draw speed, 10 mm/s; extruder filament diameter, 1.750 mm; flow adjustment, 100%; vertical shell loop count, 4 loops; thickness, 1.600 mm; and the lock H and V shell thickness were locked. SYNFASAN G3 (EMW GmbH, Diez, Germany) filters were used.

### 4.7. CPZ Content Determination

LCMS was used in all experiments, based on the determination of the active ingredient concentration. LC-MS measurements were carried out on an UHPLC system (Dionex Ultimate 3000RS) coupled to a Thermo Q Exactive Orbitrap mass spectrometer (Thermo Fisher Scientific Inc., Waltham, MA, USA) with an electrospray ionization source (ESI), using a Kinetex Polar C18 100 × 3 mm × 2.6 µm × 100 Å column. The column temperature was set to 25 °C. A solvent gradient from solvent A (water + 0.1% FA) and solvent B (MeCN + 0.1% FA) at a flow rate of 0.2 mL min^−1^ was used, as follows: 0–2 min, 0% B; 2–14 min, 0–100% B; 14–15 min, 100% B; 15–16 min, 100–0% B; 16–25 min, 0% B. A 1 µL aliquot was injected for all samples.

### 4.8. Cell Culturing

RPMI cells were maintained in an RPMI culture medium in a plastic cell culture flask, supplemented with 2 mM L-glutamine, 100 mg/L gentamycin, and 10% heat-inactivated fetal bovine serum at 37 °C in a 5% CO_2_ atmosphere. The culture medium was changed twice per week. The cells were routinely maintained by regular passaging. Prior to passaging, the flask was coated with rat-tail collagen. The cells used for the cytotoxic experiments were between passage numbers 10 and 30 [32].

### 4.9. MTT Viability Assay

To evaluate the cytotoxicity of the selected excipients, an MTT (3-(4,5-dimethylthiazol-2-yl)-2,5-diphenyltetrazolium bromide) cell viability assay was performed [33]. The experiments were carried out on the RPMI 2650 cell line, which was isolated from anaplastic squamous cell carcinoma in the nasal septum. The cells were maintained by weekly passages in the RPMI 2650 culture media. For the MTT assay, the cells were seeded on a 96-well plate, with a density of 10,000 cells/well. When the cells had fully grown over the wells’ membranes, the experiment was ready to perform. First, the culture medium was removed, and then the test solutions were applied and incubated with the cells for 60 min. After 60 min, the test substances were removed and the MTT paint solution in 5 mg/mL concentrations was added to the cells. Then, the cells were incubated for 3 h. The viable cells were able to transform the water-soluble tetrazolium-bromide into a formazan precipitate. When the incubation was completed, the formazan precipitate was dissolved with isopropanol:hydrochloride acid in a ratio of 25:1. Then, the absorbance of these solutions was measured using a spectrophotometer (Fluostar Optima); it was directly proportional to the number of viable cells. Cell viability is expressed as the percentage of the untreated control.

### 4.10. Statistical Analysis

Data were handled and analyzed using Microsoft Excel 2013 and SigmaStat 4.0 (version 3.1; SPSS, Chicago, IL, USA, 2015) and are presented as the means ± SD. Comparisons of the results of the in vitro dissolution test and MTT cell viability assays were performed using one-way ANOVA and repeated-measures ANOVA, followed by Tukey or Dunnett post testing. A difference of means was regarded as significant in the case of *p* < 0.05. All experiments were carried out in quintuplicate and were repeated at least five times.

## 5. Conclusions

In the present study, we designed a drug delivery systems for CPZ solid nanoparticles for nasal application. Nasal drug delivery has several general advantages but is particularly important for the drug under consideration here because of its serious side-effect profile. Based on the results, we found that the appropriate excipients can significantly influence the quality of these dosage forms. Amphiphilic molecules have been shown to play an essential role not only in the physical stability of drug delivery systems but also in the penetration of API. The correlations between the droplet distribution and the aggregation of nanoparticles in vaporized delivery systems were also investigated. We found that nanoparticles can be reproducibly produced using our designed and implemented method. In addition to the success of our formulation steps, the safety of our formulations was of paramount importance throughout the project. The safe applicability of the formulations was analyzed using in vitro RPMI cell viability assays. The performed studies demonstrated the safety of the drug delivery systems. The development of a customized device was required to deliver the solid nanocarriers into the nose. FDM technology was shown to be a suitable method for the development of an experimental device suitable for human application. Our results demonstrated the applicability of nanocarriers and the utility of their subclassification for nasal CPZ delivery. Our research results might provide useful data for further studies.

## Figures and Tables

**Figure 1 molecules-28-04406-f001:**
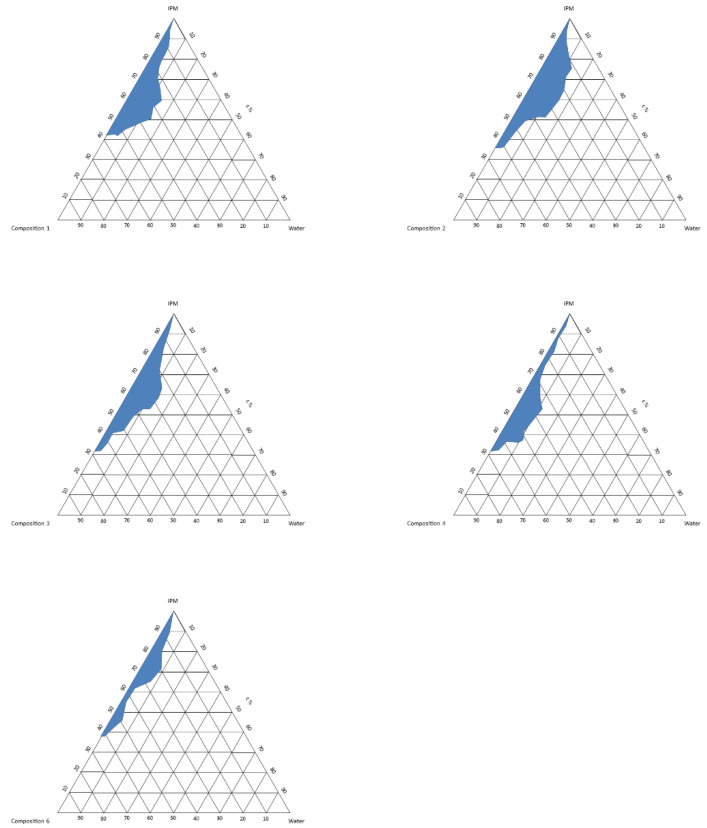
Pseudoternary phase diagram of the produced compositions; the shaded areas show the zones of the self-emulsifying system.

**Figure 2 molecules-28-04406-f002:**
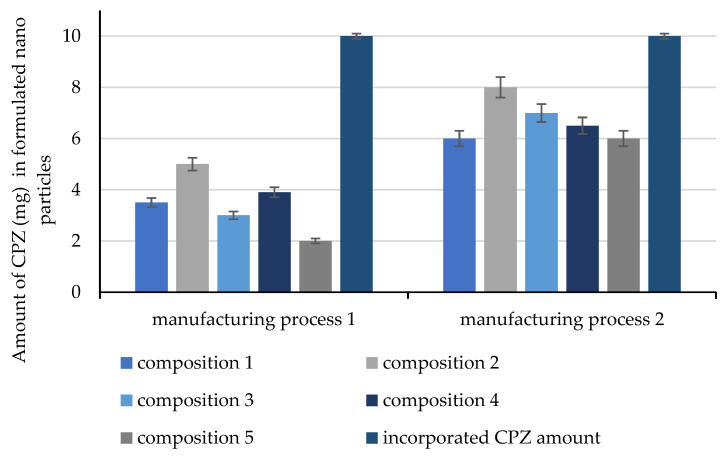
Evaluated amount of CPZ (mg) in 20 mg nanoparticles formulated by manufacturing processes 1–2. Each data point represents the mean ± SD, n = 5.

**Figure 3 molecules-28-04406-f003:**
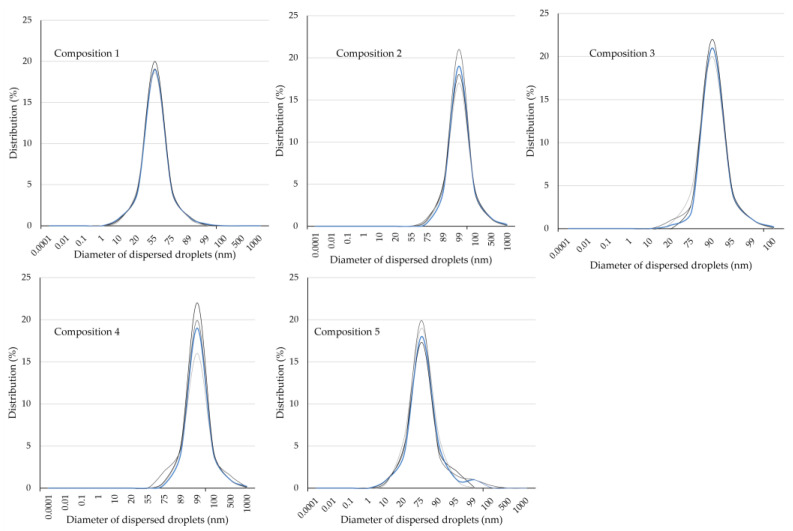
Size and size distribution of the dispersed compositions (1–5) evaluated using a Malvern ZSP nano zetasizer. (SD ± 5 nm *n* = 5).

**Figure 4 molecules-28-04406-f004:**
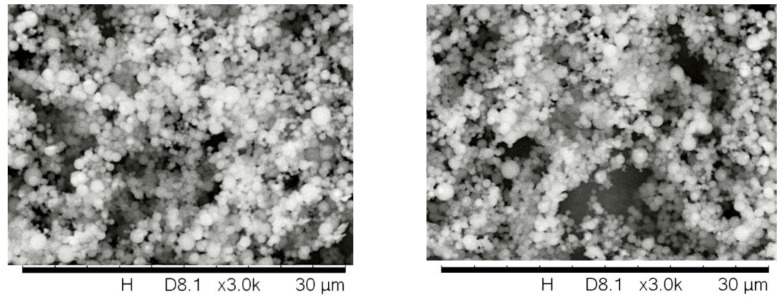
SEM images of solid nanoparticles (1–5). The measurements confirmed that the compositions take the form of heterogeneous dispersed systems in a 10–900 nm ratio.

**Figure 5 molecules-28-04406-f005:**
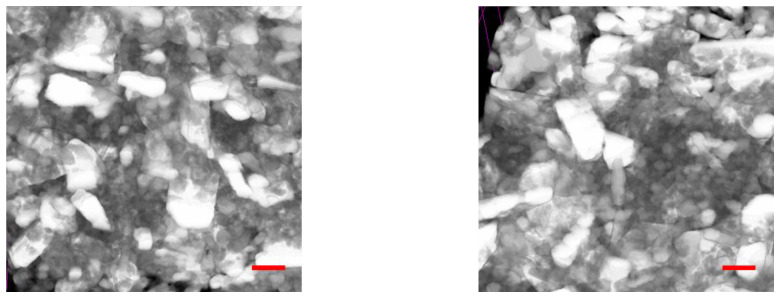
Micro CT images of solid nanoparticles in primary containers (1–5). The red scale bar in the pictures represents 500 nm.

**Figure 6 molecules-28-04406-f006:**
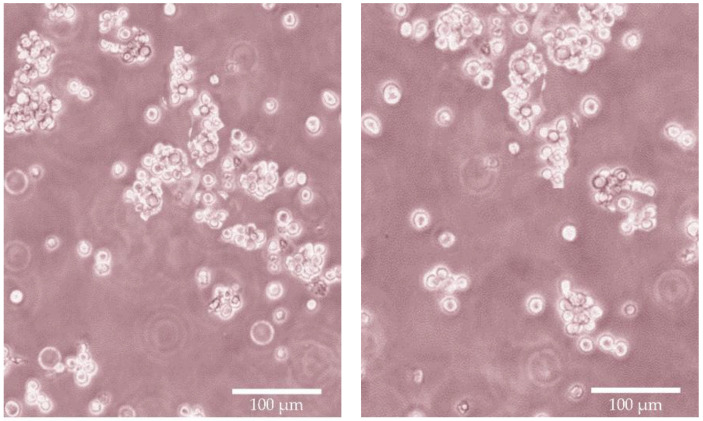
Optical microscopic images of RPMI cells before the MTT cell viability assay; 40× magnification, scale bar represents 100 µm.

**Figure 7 molecules-28-04406-f007:**
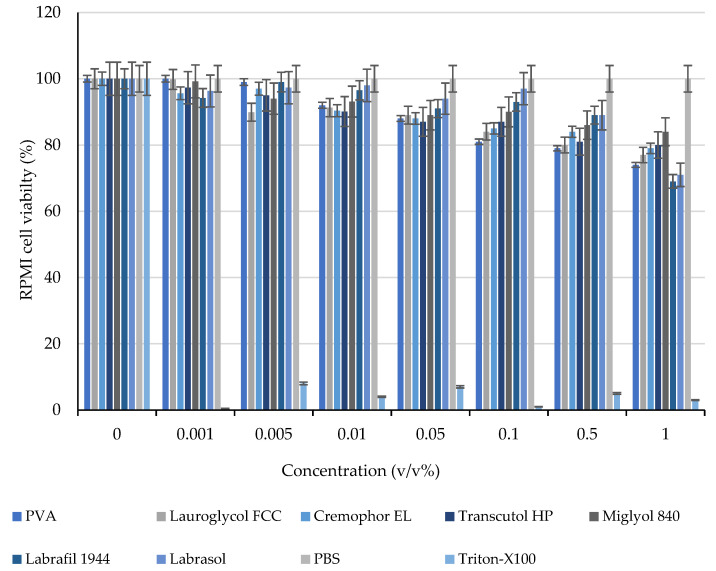
Effect of excipients on RPMI cell viability, evaluated with an MTT cell viability assay. Each data point represents the mean ± SD, *n* = 5.

**Figure 8 molecules-28-04406-f008:**
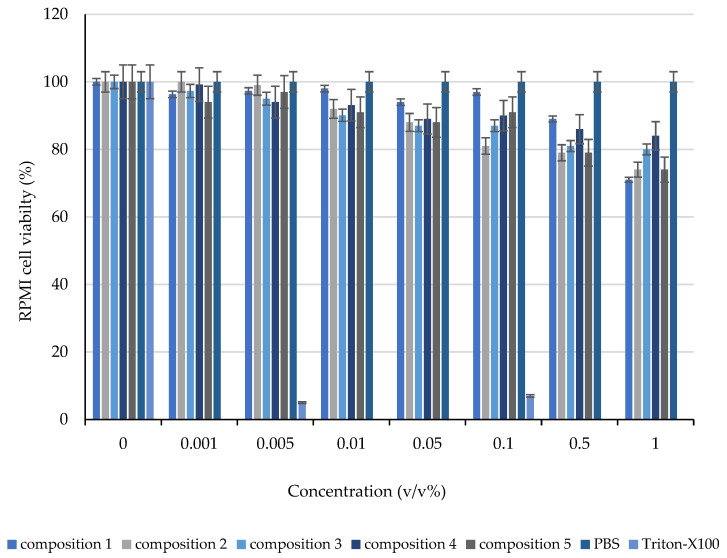
Effect of the compositions on RPMI cell viability, evaluated with an MTT cell viability assay. Each data point represents the mean ± SD, *n* = 5.

**Figure 9 molecules-28-04406-f009:**
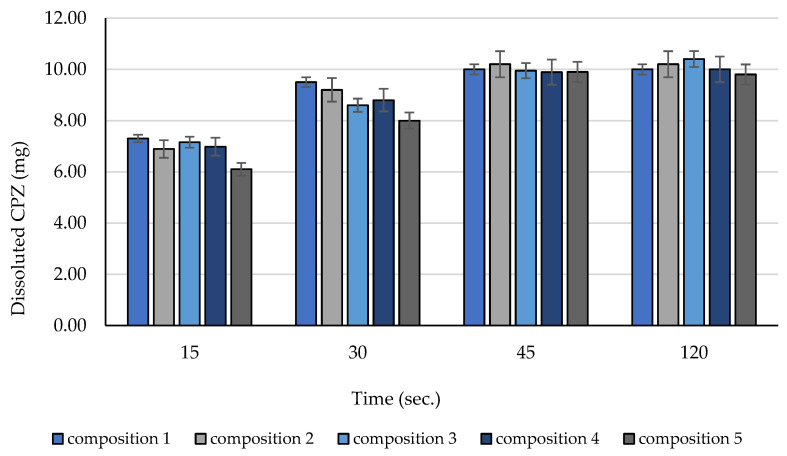
Amount of dissolved CPZ after 15, 30, 45, and 120 s. Each data point represents the mean ± SD, *n* = 5.

**Table 1 molecules-28-04406-t001:** Tested compositions of CPZ and selected excipients from the pseudoternary phase diagram. Each 1 mL composition contained 2.0 mg CPZ.

Composition	Excipients	Ratio
Composition 1	Lauroglycol FCC	1:1:1
	Cremophor EL	
	Transcutol HP	
Composition 2	Miglyol 840	1:1:0.3:1
	Cremophor EL	
	DMSO	
	Glycerol	
Composition 3	Labrafil 1944	1:1:1:1
	Labrasol	
	Cremophor EL	
	Propylene Glycol	
Composition 4	Labrasol	1:1:1:0.3:1
	Transcutol	
	Cremophor EL	
	DMSO	
	Glycerol	
Composition 5	Lauroglycol 90	1:1:1
	Cremophor EL	
	Transcutol HP	

## Data Availability

Data available on request due to restrictions. The data presented in this study are available on request from the corresponding author.

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
