# Peer review of "Development and Evaluation of an FDM Printed Nasal Device for CPZ Solid Nanoparticles"

_molecules, 2023, doi:10.3390/molecules28114406_

Round 1
Reviewer 1 Report
Dear Authors,
I studied your manuscript entitled "Development and evaluation of FDM printed nasal device for CPZ solid nanoparticles". This paper comprises interesting results that certainly deserve publication. I recommend a minor revision before further consideration for publication in the Molecules.
1) From my point of view, the abstract should be summarized. In addition, the quality of the abstract and conclusion should be improved by inserting key results of the work. It would be helpful if these sections contained some more quantitative data.
2) This manuscript has a phenomenological style, observing a result and explaining it with statements. It would be helpful if you conducted more analysis based on published research.
3) The recent literature review should be summarized for benchmarking purposes and discussed in detail with your research findings.
4) Figure 4: It is suggested that the figures with higher quality and resolution would be replaced.
5) Figure 5: scale bars are needed.
6) English language needs some polishing since some terms are vague.
English language needs some polishing since some terms are vague.
Author Response
Dear Reviewer,
First of all, we would like to express or sincere appreciation for the accurate critical review of our manuscript titled Development and evaluation of FDM printed nasal device for CPZ solid nanoparticles. We appreciate the time and effort that you have dedicated to providing your valuable feedback on our manuscript. We are grateful for the insightful comments on our paper. We have been able to incorporate changes to reflect all of the suggestions provided. We have listed the changes of the manuscript. Here is a point-by-point response to your comments and concerns.
Comment 1: From my point of view, the abstract should be summarized. In addition, the quality of the abstract and conclusion should be improved by inserting key results of the work. It would be helpful if these sections contained some more quantitative data.
Thank you very much for your comment. The manuscript has been corrected according to your kind suggestions. Moreover, the Journal has provided the opportunity to present a graphical abstract with an impressive figure summarising the whole research. This graphical abstract was created and uploaded to the MDPI website when the revised manuscript was uploaded. We also attached the graphical abstract directly to our response. We hope you and all the researchers who read our work will find it illustrative and informative enough.
Comment 2: This manuscript has a phenomenological style, observing a result and explaining it with statements. It would be helpful if you conducted more analysis based on published research.
Thank you very much for your comment. Based on your suggestion, we have further reviewed the relevant literature. We have also reviewed the list of references to the literature we used for our general findings.
Comment 3: The recent literature review should be summarized for benchmarking purposes and discussed in detail with your research findings.
During the revision, we have tried to review recent literature in depth. We have tried to improve the manuscript based on the suggestions. We hope that by making changes we have achieved a better quality and more understandable work.
Comment 4: Figure 4: It is suggested that the figures with higher quality and resolution would be replaced.
Thank you very much for your comment. When the SEM images were taken, the equipment's own software showed us the images in very high resolution and sharpness. We were very disappointed when we realised that the image quality in the manuscript formats was nowhere near the quality we would have expected. We have tried to improve the quality of the images as much as we could now. And when the research is published, we will be able to make the original high-resolution images available to all interested parties.
Comment 5: Figure 5: scale bars are needed.
Thank you very much for your comment. The figure 5 has been modified based on your suggestion.
Comment 6: English language needs some polishing since some terms are vague.
The manuscript was checked and corrected for English accuracy with the help of a native English-speaking expert. We hope the changes will further improve the clarity and quality of the manuscript.
Thank you again for your kind cooperation in the evaluation of our manuscript. We hope that with the changes we have made, you will find our article suitable and support the publication of this work in Molecules.
Yours sincerely:

Reviewer 2 Report
The following points need to revise by the authors in the revised version of the manuscript.
1. Draw a concept figure that tells your article's whole story at a glance.
2. For Figure 4, please include high-resolution images of the particles. These images are not so clear, so increase the resolution and magnification and include them in the revised manuscript.
3. In all cell viability results include cells images in all three figures 6,7, and 8.
Dear Editor
Thanks for giving me the opportunity to review this article, the article needs to be revised well before consideration for acceptance.
Thanks
Author Response
Dear Reviewer,
First of all, we would like to express or sincere appreciation for the accurate critical review of our manuscript titled Development and evaluation of FDM printed nasal device for CPZ solid nanoparticles. We appreciate the time and effort that you have dedicated to providing your valuable feedback on our manuscript. We are grateful for the insightful comments on our paper. We have been able to incorporate changes to reflect all of the suggestions provided. We have listed the changes of the manuscript. Here is a point-by-point response to your comments and concerns.
Comment 1:Draw a concept figure that tells your article's whole story at a glance.
Thank you very much for your comment. The Journal has provided the opportunity to present a graphical abstract with an impressive figure summarising the whole research. This graphical abstract was created and uploaded to the MDPI website when the revised manuscript was uploaded. We hope you and all the researchers who read our work will find it illustrative and informative enough.
Comment 2: For Figure 4, please include high-resolution images of the particles. These images are not so clear, so increase the resolution and magnification and include them in the revised manuscript.
Thank you very much for your comment. When the SEM images were taken, the equipment's own software showed us the images in very high resolution and sharpness. We were very disappointed when we realised that the image quality in the manuscript formats was nowhere near the quality we would have expected. We have tried to improve the quality of the images as much as we could. And when the research is published, we will be able to make the original high-resolution images available to all interested parties.
Comment 3: In all cell viability results include cells images in all three figures 6,7, and 8.
Thank you for your comment. We have added microscopic images of cells as an index image to our figures related to our cell viability studies.
The manuscript was checked and corrected for English accuracy with the help of a native English-speaking expert. We hope the changes will further improve the clarity and quality of the manuscript.
We hope that you will find our answers to the questions you have asked satisfactory and that, with the corrections we have made, you will find our manuscript suitable for publication in the paper.
Thank you for your support,
Sincerely

Round 2
Reviewer 2 Report
1. Images of Particles are very poor quality you must have to replace them with very high-quality images.
2. Cells images you used are the same and attached high-resolution Cells images for each group in large size in a separate panel just refer to other paper on how people show the cells images
3. Micro CT images are also not clear relace with high-resolution images
NA
Author Response
Dear Reviewer,
First of all, we would like to express or sincere appreciation for your valuable contribution in the quality improvement of our work. We are grateful for your insightful comments. We have been able to incorporate changes to reflect all the suggestions provided. Our responses to your suggestions had been summarized in the following points.
Comment 1. Images of Particles are very poor quality you must have to replace them with very high-quality images.
Thank you very much for your valuable comment. Our colleague who took SEM images made great efforts to improve the quality of the images. We have inserted these images in the revised manuscript. However, we must note that due to publication process (file format), the physical resolution of the images decreased, which worsened the quality. It was not possible to perform surface gold vaporization on these samples, nor was it possible to make better images with our equipment (Hitachi 3030) by taking repeated records. We tried to digitally sharpen the images. Please find these images below. This procedure improved the image quality.
Our research group wanted to be informed about the required quality of published SEM in connection with similar systems. We analyzed the relevant literature. Please find attached some of these references.
We hope that taking all these circumstances into account and you will support the acceptance of our images.
References:
Tumkur, P.P.; Gunasekaran, N.K.; Lamani, B.R.; Nazario Bayon, N.; Prabhakaran, K.; Hall, J.C.; Ramesh, G.T. Cerium Oxide Nanoparticles: Synthesis and Characterization for Biosafe Applications. Nanomanufacturing 2021, 1, 176-189. https://doi.org/10.3390/nanomanufacturing1030013
Bonaccorso, A.; Pellitteri, R.; Ruozi, B.; Puglia, C.; Santonocito, D.; Pignatello, R.; Musumeci, T. Curcumin Loaded Polymeric vs. Lipid Nanoparticles: Antioxidant Effect on Normal and Hypoxic Olfactory Ensheathing Cells. Nanomaterials 2021, 11, 159. https://doi.org/10.3390/nano11010159
Qinyuan Qiu, Heming Liu, Yongshan Qin, Chunguang Ren, Jian Song Efficiency enhancement of perovskite solar cells basedon Al2O3-passivated nano-nickel oxide film Journal of Materials Science 2020 55(28)
https://doi.org/10.1007/s10853-020-04965-0
Mayekar J, Dhar V, Radha S, To Study the Role of Temperature and Sodium Hydroxide Concentration in the Synthesis of Zinc Oxide 2013 Nanoparticles. 3. 11,
ISSN 2250-3153
Attached reply document contain our original SEM figures and the improved images.
- Cells images you used are the same and attached high-resolution Cells images for each group in large size in a separate panel just refer to other paper on how people show the cells images
Thank you for your comment. The optical microscopic images of the RPMI cells were inserted in a separate figure in the corrected manuscript according to your suggestion. The photograph of the cells was taken on the fifth day after the first passage. No microscopic determinations were made during the tests. Cell viability had been determined according to the enzymatic reduction of 3-[4,5-dimethylthiazole-2-yl]-2,5-diphenyltetrazolium bromide (MTT) to MTT-formazan is catalyzed by mitochondrial succinate dehydrogenase. During the viability tests, the morphology of the cells was not examined, only their mitochondrial activity was monitored.
- Micro CT images are also not clear relace with high-resolution images
Thank you for your comment. Images had been replaced with new images in the revised manuscript.
We would like to express again our appreciation for your effort to improve our manuscript. We hope our answers are acceptable and you will be so kind to support the publication of our work.
Sincerely,
Zoltán Ujhelyi PharmD PhD

Round 3
Reviewer 2 Report
NA
NA